# Comparative Assessment of In Vitro and In Vivo Biodegradation of Mg-1Ca Magnesium Alloys for Orthopedic Applications

**DOI:** 10.3390/ma14010084

**Published:** 2020-12-26

**Authors:** Iulian Antoniac, Răzvan Adam, Ana Biță, Marian Miculescu, Octavian Trante, Ionuț Mircea Petrescu, Mark Pogărășteanu

**Affiliations:** 1Faculty of Materials Science and Engineering, University Politehnica of Bucharest, 313 Splaiul Independentei, 060042 Bucharest, Romania; iulian.antoniac@upb.ro (I.A.); ai.blajan@gmail.com (A.B.); octavian.trante@upb.ro (O.T.); ionut.petrescu@upb.ro (I.M.P.); 2Orthopedic Department, Clinical University Emergency Hospital Elias, 17 Mărăști Blvd., 011461 Bucharest, Romania; 3Orthopedics-Traumatology Department, Carol Davila University of Medicine and Pharmacy Bucharest, 3-7 Dionisie Lupu Str., 020022 Bucharest, Romania; mark.pogarasteanu@gmail.com

**Keywords:** Mg-Ca alloy, SEM/EDS, In vitro, In vivo, biocompatibility, corrosion rate

## Abstract

Use of magnesium implants is a new trend in orthopedic research because it has several important properties that recommend it as an excellent resorbable biomaterial for implants. In this study, the corrosion rate and behavior of magnesium alloys during the biodegradation process were determined by in vitro assays, evolution of hydrogen release, and weight loss, and further by in vivo assays (implantation in rabbits’ bone and muscle tissue). In these tests, we also used imaging assessments and histological examination of different tissue types near explants. In our study, we analyzed the Mg-1Ca alloy and all the hypotheses regarding the toxic effects found in in vitro studies from the literature and those from this in vitro study were rejected by the data obtained by the in vivo study. Thus, the Mg-1Ca alloy represents a promising solution for orthopedic surgery at the present time, being able to find applicability in the small bones: hand or foot.

## 1. Introduction

Bioabsorbable metal osteosynthesis materials are the next step in orthopedic surgery. These have the advantage of superior mechanical properties compared to polymers, being made up from elements already present in the human body [1]. The main requirements of these materials are biocompatibility and biofunctionality, with no rejection reaction by the surrounding tissues, and it should also assist the bone repair [2]. Furthermore, they require no toxic reactions when resorption occurs, bone adhesion, and to allow bone cell proliferation (osteoconductive) and resorption by biodegradation and bone remodeling.

The use of resorbable materials in orthopedic surgery has the advantage of avoiding further ablation surgery, reducing patient’s risks and stress and reducing the costs for the medical system [3]. At the same time, by resorption, it will reduce the contact stress of the bone/implant (stress shielding effect) and allow the transfer of the load capacity gradually from implant to the bone, reducing the risk of post-implant fracture, as in the case of conventional, non-absorbable implants.

Moreover, the use of magnesium implants is a new trend in orthopedic research because it has several important properties that recommend it as an excellent resorbable biomaterial for implants [4]. Mg-based implants show mechanical properties similar to the mechanical values of bone. They have a density of 1.7–2.0 g/cm^3^, close to natural bone density (1.8–2.1 g/cm^3^) and an elastic modulus of approximately 45 GPa, close to that of natural bone (10–40 GPa) [3,5]. Comparatively, usual metallic implant’s elastic modulus is 100–200 GPa [5].

The main drawback of Mg use in medical contexts is its rapid biodegradation, having an increased rate of corrosion in the body at a physiological pH of 7.4 to 7.6 [6,7]. Alloying is a good way to diminish these disadvantages. 

The elements released from the alloying implant must not be toxic for human tissues. Mg-Ca alloys respect these requirements, being composed of two elements naturally present in the human body. Magnesium is an important element for the human body, and calcium is an important component of bone tissue and is an essential element in biochemical processes [8,9]. According to the phase diagrams, the solubility of Ca in Mg is 1.34 wt% [4,10]. Ca’s low density (1.55 g/cm^3^) makes Mg-Ca alloys achieve a density similar to bone. At the same time, magnesium has the potential to improve the incorporation of calcium into bone [11,12]. 

Mg-Ca alloys have a binary microstructure, consisting of a primary phase, α Mg, and a eutectic secondary phase, with lamellar structure, composed of Mg_2_Ca [13]. From the electrochemical behavior point of view, the Mg_2_Ca phase is more active than the α Mg phase, thus having a cathodic function with respect to magnesium, this difference leading to the formation of a galvanic circuit. The corrosion process of Mg-Ca alloys is the result of this phenomena, due to the difference of electrochemical potential between the primary phase α Mg and the secondary phase Mg_2_Ca. Mg-Ca alloys undergo the next corrosion process (Figure 1) in simulated biological fluid (SBF):

(a) Galvanic corrosion of Mg and Mg_2_Ca phase; (b) partially protective layer covering the surface of the Mg-Ca alloy; (c) absorption of chlorine ions for the conversion of Mg(OH)_2_ to MgCl_2_; (d) hydroxyapatite formation by consuming Ca^2+^ and PO_4_^3−^ ions; (e) particulate residues decay emerging from the substrate volume.

The amount of calcium in the alloy will influence its microstructure and, implicitly, the mechanical properties and its corrosion resistance. The addition of calcium will lead to refining of the phase grains, forming thermally stable metallic phases and thus improving the mechanical qualities and corrosion resistance of the alloy. However, with the increase of calcium content, the Mg_2_Ca phase will precipitate to the grain boundary in increased quantities, having the effect of decreasing both mechanical properties and corrosion resistance [14].

The results presented in the literature, obtained on alloys with low calcium concentrations, 1, 2, and 3%, were better than those with high concentrations (5–20% Ca), both in terms of mechanical properties and corrosion rate [10,15]. Of these, Mg-Ca 1% alloys were considered to be optimal for orthopedic implants [16]. Furthermore, another method of improving the mechanical properties of Mg-Ca alloys is the secondary mechanical processing. The tensile strength of the alloy has recorded, initially, a value of 71.38 ± 3.01 MPa. After hot rolling, it reached a value of 166.7 ± 3.01 MPa, and after hot extrusion it was improved up to a value of 239.63 ± 7.21 MPa [16].

Mechanical processing, for grain refining, will also lead to an improvement of the alloy’s corrosion rate. The diminished corrosion rate of the Mg-1Ca alloy, processed by hot extrusion, has been expressed by a low hydrogen emission rate, from 35 mL/cm^2^ to 10 mL/cm^2^ [16].

A high corrosion rate, faster than bone healing rate, leading to a rapid loss of mechanical capabilities, faster than the bone growth rate was the starting point of this study. Further, a high hydrogen releasing rate leads to gas bubble formation into the surrounding tissues, leading to tissue necrosis, surgical incision dehiscence [17], and risk of gas embolization in vital organs [18].

A high corrosion rate of magnesium alloys will lead to the increase of the pH in the tissues surrounding the implant, thus producing an alkalizing effect of peri-implant tissues, with cytotoxic effect. According to ISO 10993-5:2009, a reduction of cell viability more than 30% is considered a cytotoxic effect [19,20,21]. As a result of the corrosion process of magnesium alloys, a layer of oxidative products will form on their surface, with a partial protection role, that is stable at alkaline pH. In the case of physiological pH (7.4), it was believed that this layer would be dissolved, thus leading to an increased corrosion rate [22].

Some authors [23,24] point out that chlorides from the human body will interact with the protective layer of Mg(OH)_2_ forming magnesium chloride, soluble in water. Moreover, enzymes and amino acids will produce complexes with an inhibitory effect on inorganic salts deposited on the surface of the magnesium alloy, thus accelerating the corrosion process [25,26,27]. On the other hand, there are some opinions [22,28] that state that HCO^3−^ and HPO_4_^2−^ ions have a protective effect, contributing to the temporary formation of a hydroxyapatite layer on the surface of the alloy, and also the proteins of the human body will adhere to the surface of the implant forming a protective layer with a cathodic reaction inhibiting effect, reducing the corrosion rate. At the same time, the protein layer will increase the cell adhesion to the surface of the implant having a positive effect on its biocompatibility [22].

Data from other published studies hypothesize that a high corrosion rate, faster than bone healing rate, will lead to a rapid loss of mechanical capabilities, faster than the bone growth rate. A high hydrogen evolution rate will lead to the formation of gas bubbles into the surrounding tissues, leading to tissue necrosis, surgical incision dehiscence [7], and risk of gas embolization in vital organs [8]. 

These studies, by their obtained data, lead to the conclusion that magnesium alloys have a fast corrosion rate, leading to early loss of mechanical properties, and to local and systemic toxic phenomena limiting the use of these alloys in medical applications.

These opposing conclusions are common in the literature, thus, it is considered that the concordance between the values obtained in in vitro and in vivo tests is reduced, in vitro tests having no good predictability for in vivo tests [29].

Starting from these hypotheses, we try in this paper to evaluate an Mg-Ca alloy, in both In vitro and In vivo tests, to demonstrate its qualities as a potential material for medical applications.

This work aims to assess the corrosion rate and behavior of magnesium alloys during the biodegradation process, which were initially determined by In vitro assays, by the evolution of hydrogen release and weight loss, and further by In vivo assays (implantation followed by imaging assessments and histological examination on different tissue types).

## 2. Materials and Methods

### 2.1. Samples Materials and Preparation Method

The Mg-Ca alloy raw material was obtained using compact magnesium (99.7% purity) and granules of calcium (99.8% purity) in a rocker crucible furnace with sulphur hexafluoride (to protect against oxidation). Calcium granules were added to the melt at the temperature of 680 °C, and after mixing, the load was poured into a metal plaster pre-heated at 250 °C. A CT-AL-1.1 electric crucible furnace (Protech, Zhengzhou, China) with a graphite crucible with kantal resistors, automatic temperature control (±5 °C), and instant thermocouple were used. The ingots were obtained by casting into a permanent mold using a mild steel crucible. After solidification and cooling of the cast, the bar was trimmed and cleaned. The prepared bulk Mg-Ca alloy was cut into shaped pieces and the sample’s surface was polished and cleaned using metallographic techniques for magnesium alloys, described previously, without any special surface pre-treatment.

SEM-EDS results obtained in 3 different areas of the control sample for the prepared Mg-Ca alloy are presented in Table 1.

### 2.2. In Vitro Tests

#### 2.2.1. Corrosion Rate Assessment (Weight Loss Method)

The experiments were performed by immersing the Mg-1Ca alloy samples in a solution of simulated biological fluid, SBF, and maintaining them for fixed periods of time. We used 3 identical, parallelepiped samples, with dimensions of 1.2 cm/1.2 cm/0.2 cm. The exposed surface of the sample was 3.84 cm^2^. The density of the alloy was 1.78 g/cm^3^. The weight of the sample before immersion was 0.512 g and was measured using an analytical balance. The samples were weighed before immersion and immediately after washing the solution and cleaning corrosion products, using a chromic acid solution. From the resulting data, we calculated the In vitro corrosion rate. To mimic the physiological conditions as accurately as possible, the solution was buffered to a pH of 7.4. To maintain the samples and corrosion medium at 37 ±1 °C, an Immersion Bath Circulator Lab Tech, model LCB-11D bath thermostat (LabTech, Gyeonggi-do, Korea) was used. Every sample was introduced into 45 mL corrosion medium for 72, 120, 168, 240, and 288 h.

The mass loss (M.L.) was calculated using the following formula:

M.L. = ((m_0_ − m_t_)/m_0_) × 100
(1)
where:m_0_—mass of the sample before immersion;m_t_—sample mass, at different time intervals, after the removal of the corrosion products.

The corrosion rate, derived from weight loss, was determined using the following formula:
CR = (8.76 × 104 × ΔW)/A × t × ρ,
(2)
where:CR—corrosion rate (mm/year); ΔW—weight difference, before and after immersion (g); A—the area of the initial surface exposed to corrosion (cm^2^); t—immersion time (h);ρ—density of the alloy (g/cm^3^).

The pH of corrosion media plays an important role in the evolution of the corrosion process. Within the biodegradation process of the Mg-1Ca alloy in SBF, alkalization of the environment in the immediate vicinity of the implant will occur, leading to enhanced formation and deposition of calcium phosphate compounds, and thus the implant will have a thicker protective layer. In addition to its role in reducing the corrosion rate, hydroxyapatite (HA) will stimulate bone formation and thus the implant’s osteointegration. The minimum pH value at which the Mg(OH)_2_ and HA layer is stable is 10.4. Lower values close to the physiological pH will lead to the disintegration of this layer, accelerating the corrosion process [30]. The test solution’s pH determination was carried out at a temperature of 37 ± 1 °C in a thermostatic bath, using a pH metermodel HI2210 produced by Hanna Instruments (Woonsocket, Rhode Island).

#### 2.2.2. Corrosion Rate Assessment (Hydrogen Release Method)

This experiment was based on the fact that during the corrosion process of the Mg-1Ca alloy, hydrogen is released as a gas, a relatively simple principle. The volume of hydrogen released from the alloy during the corrosion process is proportional to the amount of the decomposed alloy. Thus, the volume of hydrogen collected is converted to material loss, according to the ratio 1 mL of hydrogen released = 0.001083 g of decomposed alloy [31]. Subsequently, the corrosion rate was calculated according to the same method as in the weight loss test, using the formula:

CR = (8.76 × 104 × ΔW)/A × t × ρ,
(3)
where:CR—corrosion rate (mm/year); ΔW—weight difference, before and after immersion (g); A—area of initial surface exposed to corrosion (cm^2^); t—immersion time (h);ρ—density of the alloy (g/cm^3^).

In these tests also, three parallelepiped samples were used, made from Mg-1Ca alloy, with the sample sizes of 1.2 cm/1.2 cm/0.2 cm, and a 3.84 cm^2^ exposed surface. The density of the alloy was 1.78 g/cm^3^.

The samples were cleaned and disinfected in acetone solution, then introduced into the gas emission measuring device.

We used the complex Dulbecco corrosion medium in order to simulate the physiological conditions in living organisms as accurately as possible and to have a point of comparison with the corrosion rate obtained by immersion in SBF. To simulate the physiological conditions from living organisms, the test was performed in a thermostatic bath (Immersion Bath Circulator LabTech, model LCB-11D), at a temperature of 37 ± 1 °C. The Dulbecco medium was buffered to a pH of 7.3.

#### 2.2.3. Cell Line, Cytotoxicity Tests

The cytotoxic effect of Mg-1Ca alloy on cell viability was made using the AlamarBlue^®^ test method (Cell Titer-Blue Cell Viability Assay, Promega), an analysis based on the redox reaction using the fluorescent dye, resazurin. Fluorescence was analyzed using the Micro-Plate Reader VICTOR3 plate reader (PerkinElmer, Shelton, CT, USA).

Cell cultures, human osteogenic sarcoma cells (SaOs-2), purchased from the American Type Tissue Collection (ATCC HTB 85, Manassas, VA, USA), were used. We performed an indirect test in the cell culture medium with an added extract made from Mg-1Ca alloy on Dulbecco culture medium supplemented with 10% FBS (fetal bovine serum) at 37 °C. We used 2 extracts, at 24 and 72 h, with a negative control, high density polyethylene (HDPE), and a positive control (latex). The test was started when we obtained an 80% cell confluence in culture medium (Figure 1). We used an alloy sample weighing 0.4905 g immersed in 2.452 mL culture medium. The extracts collected at 24 h and at 72 h were added to the cell cultures, practically replacing the culture medium. The viability of the cells was evaluated at 1, 2, 3, 7, and 12 days. Evaluation at short time exposure 24 h up to 3 days, provides information on the direct toxic effect (cell necrosis) while the evaluation at 7 and 12 days (long exposure time) provides information regarding inhibition of cell proliferation.

#### 2.2.4. SEM (Scanning Electron Microscopy) and EDS (Energy Dispersive Spectrometry) Examination

SEM is an efficient evaluating method of the way in which the magnesium alloy implant degrades, and of the characterization of its surface structure both before and especially after implantation [32].

We can obtain information on the grains’ dimensions and their distribution at the grain boundaries, these playing an important role in determining the corrosion behavior of the studied alloy. Energy dispersive spectrometry (EDS) provides data about the elements present on the implant surface as a result of corrosion but also as a result of the biointerference of the implant. In this way, we can evaluate the osteoconductive capacity of magnesium alloy implants by determining, for example, whether elements of the hydroxyapatite component appear on its surface [33]. Samples of Mg-1Ca alloy were evaluated by these methods after immersion in Dulbecco corrosion medium.

#### 2.2.5. In Vivo Study

The study was carried out with approval of the Ethics Committee for Scientific Research in University of Medicine and Pharmacy “Carol Davila” Bucharest, no. 92/20.04.2016, code PO-35-F-03. Animal care and clinical evaluation were performed according to FELASA (Federation of European Laboratory Animal Science Associations) guidelines and recommendations (121ld).

Three domestic *Oryctolagus Cuniculus* rabbits, adults with a minimum weight of 3.5 kg, were used and numbered from 1 to 3. Mg-1Ca alloy samples were implanted into bone tissue (greater femoral trochanter) and in muscular tissue (thigh musculature), with a 6-week follow up. The same type of implant as in the In vitro studies was used (Figure 2).

**Pre-operative procedures:** Lower limb radiographs were made, in two incidents, anterior-posterior and laterally (Figure 3). The animals were dewormed using Ivermectin, at a dose of 0.1 mL/kgC, subcutaneously (SC). Pre-operative antibiotic therapy was used (one dose of Enrofloxacin (Boytril 5%) at a dose of 10 mg/kgC, subcutaneously).

**Surgical procedures:** The implants were chemically sterilized by ethylene oxide sterilization method, thus avoiding damage to the alloy. The type of anesthesia chosen was veterinary mask inhalation anesthesia, using Sevoflurane. For induction, Ketamine 20 mg/kg C, subcutaneously (SC), was used. Throughout the surgery, we monitored the rabbits’ cardiac activity and respiratory movements. A lateral approach was used at 2 cm, in the 1/3 proximal thigh, centered on the bony relief of the greater femoral trochanter. The lateral cortex was drilled at the base of the greater trochanter using a 3.2 mm diameter drill, in the femoral neck direction (Figure 4a). In the created orifice, the implant was inserted by hammering (Figure 4b). By lateral approach (longitudinal, 3 cm long, middle third thigh), the intersection between the vastus lateralis muscle and the intermediate vastus muscle (anterior head) was identified, and in this space an Mg-1Ca alloy implant was introduced (Figure 4c).

**Post-operative procedures:** Prophylactic antibiotic therapy was used, two days post-operatively (Enrofloxacin (Boytril 5%) at a dose of 5 mg/kgC/12 h, SC. Postoperative pain was controlled using Meloxicam at 0.2 mg/kgC/24 h, SC.

Two incidents radiographs, anterior-posterior and lateral, were made, with the same radiological device and the same technical data as in the case of pre-operative radiographs. Radiological follow up was necessary to highlight the resorption of implants, the quality and integrity of the bone, and evolution of the gas (hydrogen) bubbles. 

Euthanasia was performed 6 weeks after implantation. In order to eliminate the suffering of the animals, we chose anesthesia as an initial procedure, performed by SC Ketamine injection at a dose of 20 mg/KgC. After this, euthanasia was performed by injecting 1 mL T61 (Etambutramide + Mebezonium Iodide + Tetracaine hydrochloric) intravenously (IV).

**Post-euthanasia procedures:** Mg-1Ca alloy implants ablation, both from bone and muscle tissue. The implants were subjected to corrosion rate evaluation, using weight loss method, as in the in vitro tests. At the same time, we calculated the sample’s percentage mass loss using the following formula:

M.L. = ((m_f_ − m_0_)/m_0_) × 100
(4)
where m_0_—sample mass before immersion and m_f_—final sample mass at the time of ablation.

Implants were also examined by SEM and EDS. For the above mentioned analyses, we used a QUANTA INSPECT F scanning electron microscope (FEI Company Hillsboro, OR, USA) equipped with a field emission gun (FEG) with 1.2 nm resolution and energy dispersive X-ray spectrometer (EDS) (FEI Company, Hillsboro, OR, USA) with an MnK resolution of 133 eV. Tissue samples from muscle and bone, located in the contact area with the implant were preleved and were sent to the laboratory for histological examination. Bone tissue was harvested from the distal femur, away from the implantation area, as control sample. Histological examination aimed to highlight cell viability, morphology of bone or muscle cells in the immediate vicinity of the Mg-1Ca alloy implant, and the presence of gas bubbles. Muscle tissue samples were fixed in paraffin block. Using a microtome, 4 μm thick sections were cut. The sections were fixed on examination blades and colored. The staining method was hematoxylin-eosin. The slides were analyzed using a Nikon Eclipse E200 optical microscope (Nikon Corporation, Minato City, Tokyo, Japan), connected to a computer for the analysis of the obtained images. Bone tissue samples were decalcified in 10% EDTA solution. After decalcification, 4 μm thick sections were fixed in paraffin blocks, from which microtomes were cut. The sections were fixed on examination blades and colored. The staining method was hematoxylin-eosin, Van Gieson, and Tricrome Masson. The slides were analyzed using the same type of optical microscope.

## 3. Results and Discussion

### 3.1. In Vitro Tests

#### 3.1.1. Mass Loss Method, in SBF

For the mass loss, we notice a marked increase towards day 7, 168 h of immersion, with a stabilization, slower rate, at day 12, 288 h of immersion (Figure 5).

A similar behavior was observed in the case of corrosion environment alkalization (Figure 6).

Regarding the corrosion rate in SBF, a constant increase was observed during the first week, with a maximum at 72 h of immersion (Figure 7). After this time, we recorded a decrease in the corrosion rate, at 288 h of immersion, reaching values similar to the ones recorded on day 7.

#### 3.1.2. Hydrogen Release Method, in Dulbecco Medium

By converting the volume of released hydrogen, measured at the set time intervals, into daily release values, we obtained for Mg-1Ca substrates a maximum rate of hydrogen released in Dulbecco medium (Figure 8) of 3.76 mL/cm^2^/day, higher than the rate of maximum daily absorption of the human body, 2.25 mL/cm^2^/day.

Regarding the evolution of in vitro corrosion rate (Figure 9) calculated by the volume of hydrogen release, we observed that during the first 24 h there was an accelerated increase of corrosion rate of 6.93 mm/year, this gradually decreasing to day 7, 168 h of immersion at a value of 1.98. After this period, there was a light rate increase, until the 12th day, 288 h of immersion, reaching a value of 3.61 mm/year. The dynamic evolution of the corrosion rate was similar to the hydrogen release rate evolution, per day.

#### 3.1.3. Cell Viability Tests

Cell viability was significantly higher than the control sample, overall, obtaining an average viability of 77.14%. In Figure 10a, osteoblasts evolution at day 2, in 24 h extract is presented. Cell viability was significantly higher than the control sample. The average cell viability decreased compared to the 24 h extract, being 47.11%. Sinusoidal evolution of cell viability, with a big difference between maximum and minimum value, 39.52%. In Figure 10b right, osteoblasts evolution at day 7, in 72 hours’ extract is presented.

#### 3.1.4. SEM and EDS Results

Figure 11 shows the SEM image of the Mg-1Ca alloy, before immersion. Number 1 indicates the grain, and in detail, the grain limit is observed, being marked with 2. 

In Figure 12, SEM at 1, 2, 3, 7, and 12 days of immersion in SBF compared to Dulbecco complex medium are presented, and before and after the corrosion products formed on the surface of the alloy were removed. SEM investigations performed on the surface of the Mg-1Ca alloy after testing the corrosion resistance by immersion tests revealed a morphology of the surface specific to corrosion, the material showing cracks on the entire tested surface.

After removing the corrosion products from the surface of the experimental alloy Mg-1Ca, surface morphologies specific to crevasse corrosion and pitting type were highlighted.

Table 2 shows EDS results before immersion, in which we identify the preponderance of magnesium, 99%, calcium being present in a proportion of 1%. At the same time, we observed residual compounds, phosphorus, and oxygen, detected on the surface of the alloy, as well as residual gold from the sample fixing plate. Chromium presence can be attributed to the cleaning solution that was used, with a solution of chromic acid and silver nitrate (200 g L^−1^ CrO3 + 10 g L^−1^ AgNO3), and that led to the removal of Ca-P compounds (hydroxyapatite-specific compounds which are also found in bone tissue). Corrosion products are identified in the form of elements, carbon, phosphorus, oxygen, sodium, and potassium. The presence of these elements can be justified by the formation of hydroxyapatite and calcium phosphates on the surface of the alloy, during the corrosion process. 

### 3.2. In Vivo Tests

#### 3.2.1. Radiological Follow-Up

The muscle implant shows moderate changes in shape. The bone implant shows more advanced changes in shape. Transparent delimitation lines with bone tissue are not observed. Hydrogen bubbles have predominant development in the soft tissues (Figure 13).

#### 3.2.2. Histological Results

In muscle sections (Figure 14a), hydrogen bubbles released intramuscularly, with a white appearance, no. 1, muscle fibers, no. 2, normal color and viable appearance. The section shows muscle fibers nucleus, no. 3, colored with blue, a sign of cell viability. The control sample shows muscle cells, viable, no. 1 and cell nucleus, no. 2, with a similar aspect to tissue samples muscle collected from the vicinity of the implant.

Mg-1Ca alloy, in direct relationship with bone tissue of normal appearance, osteocytes, no. 1, being present in large number, in the form of disks, with nucleus at interior, no. 2, and circular arrangement of the osteons, marked with 2 in Figure 15b, sign of cell viability. The Mg-1Ca alloy, marked with 3 in the figure, is in an advanced resorption stage. The alloy is in direct contact with bone surface, adherent to it. At the alloy bone interface, marked with the number 4, no other tissues are identified, fibrous or cartilaginous. Near the bone implant interface, we observe an area of bone tissue, more intensely colored (5), a mineralization area, sign of bone activity.

To highlight the cellular activity and viability of bone tissue in the vicinity of the Mg-1Ca alloy implant, we performed sections, Tricrome Masson dye as shown in Figure 15c. The viable bone cells with intracytoplasmic nucleus, number 1, are highlighted. It can be observed that collagen fibers, colored in blue, no. 2, are arranged both around the cells and in the bone mass, sign of osteoblastic activity. The Mg-1Ca alloy, no. 4, partially resorbed, is directly adherent to the bone surface, without identifying interposing tissues at the bone-alloy interface, no. 5. In the immediate vicinity of the implant, the bone tissue is disposed in parallel lines, no. 3, while at distance, it has a normal, circular arrangement, in osteons, no. 3, this demonstrating the formation of new bone at the interface with the Mg-1Ca alloy.

Figure 15c right—bone control sample. Hematoxylin eosin dye. 10 × 4 optic magnitude, trabecular bone tissue, no. 1, viable, normal appearance, with osteocytes, as dark points. Cortical bone, no. 3, with the same characteristics as the trabecular one. Aspect similar to tissue samples bones collected from the vicinity implant from Mg-1Ca alloy.

#### 3.2.3. In Vivo Quantitative Corrosion Rate Assessment, Weight Loss Method

The obtained data show a higher corrosion rate for the bone implanted alloy compared to the one implanted in the muscles (Figure 16). The average corrosion rate value of the bone alloy is 1.34 mm/year, while for the alloy implanted in the muscles, it is 0.703 mm/year (Table 3).

The statistical analysis of the differences between the corrosion rate in SBF and the corrosion rate in bone and muscle tissue, presented below, shows a major, statistically significant difference, *p*—0.025 (Table 4).

A similar situation is observed when comparing the corrosion rate in Dulbecco environment with the corrosion rate in the bone and muscle tissue, respectively. The difference registered is major, statistically significant, *p*—0.020.

For statistical analysis of the comparison between corrosion rates (Table 5), we used the Mann–Whitney test for two independent samples, as a test for ordinary data [34]. To obtain these results, we used the statistical processing software for portable SPSS data 20.

## 4. Discussions

Complex environments, like Dulbecco, will induce a much lower corrosion rate compared to simple ones, like SBF. The composition of the environment plays an important role in corrosion rate evolution, the more the environment is similar to living organisms, the lower the corrosion rate will be. Thus, the hypothesis of in vivo corrosion rate being lower than in vitro is raised [8,35].

The pH in both corrosion environments reached high values, raising the hypothesis of in vivo cytotoxicity phenomena [36]. The hydrogen release rate is higher than the human body absorption rate, predisposing to the formation of gas bubbles.

The concentration of ions released in the culture medium influences the cell viability rate, a fact indicated by the differences recorded between the two types of extract in cell line tests.

At a reduced ion concentration in the culture medium, the Mg-1Ca alloy is included in the international standard degree of cytotoxicity (ISO 10993-5: 2009), being able to provide good cell viability. On the other hand, ions released at a high concentration of alloy (extract 72 h) equivalent to an increased corrosion rate, can induce cellular, local, or systemic toxic effects.

The corrosion rate of the Mg-1Ca alloy in SBF is very high, the alloy quickly losing its structure, being practically disintegrated at 12 days of immersion. The pH of the SBF solution, but also of the Dulbecco solution, reached too high values for physiological processes, raising the hypothesis of the occurrence of cytotoxicity phenomena in living organisms, by blocking some pH-dependent biological mechanisms. This hypothesis requires testing by in vivo studies.

The pH of the corrosion medium directly influences the corrosion process. Through the degradation process, the alloy will alkalize the environment, and this alkalization will decrease the corrosion rate, allowing the stabilization of the Mg(OH)_2_ layer and the formation of additional layers of hydroxyapatite or protein film.

The simulated body fluid, SBF, although it allows the formation of the HA layer on the surface of the alloy, will induce a fast, aggressive corrosion process with the material, and a Ph level too high to provide safety in a living organism.

The Mg-1Ca alloy, the cast form, produces hydrogen during the corrosion process, at a higher rate than the adsorption capacity of the human body. It predisposes to the formation of gas bubbles in the tissues adjacent to the implants made of this magnesium alloy.

Dulbecco-type complex media provide better protection for the Mg-1Ca alloy, with a much lower corrosion rate than SBF. This raises the hypothesis of a lower in vivo corrosion rate than in vitro. At the same time, secondary to a slower corrosion process, the alkalization, although increased, is lower than in SBF.

The corrosion environment plays an important role in the corrosion dynamics of the Mg-1Ca alloy. The more similar the environment is to the physiological environment in living organisms, the more controlled the corrosion behavior of the alloy. These observations indicate that the Mg-1Ca alloy provides a degree of safety and can be used in in vivo studies.

The concentration of magnesium and calcium ions in the culture medium and corrosion products influences the cell viability rate, a fact indicated by the differences registered between the two types of extract. The Mg-1Ca alloy can provide good cell viability, with reduced cytotoxic effects, and little influence on the cell proliferation process. At a low ion concentration in the culture medium, the Mg-1Ca alloy is in line with the international standard on the degree of cytotoxicity.

The results obtained using the 72-h extract lead to the hypothesis that ions released in an increased concentration of the alloy, equivalent to an increased corrosion rate, can induce toxic cellular, local, or systemic effects. The data obtained in the in vitro study on the effect of the degradation process of the Mg-1Ca alloy on cell viability are encouraging, justifying further research by in vivo studies.

In vivo cell viability studies are required to certify the results obtained in vitro and to verify the hypothesis of local and systemic cytotoxic effect induced by a rapid corrosion process, with massive release of ions into the environment.

Furthermore, the Mg-1Ca alloy has the ability to biodegrade, disintegrating over time into compounds that are resorbed by the body. During resorption, it does not induce local toxic phenomena and it does not change the structure or viability of the tissues in which it is implanted, muscle or bone. Degradation products released systemically during the resorption process of Mg-1Ca alloy, or hydrogen released in the form of gas, do not lead to pathological structural changes in the mentioned tissues. The resorption process does not significantly affect the general clinical condition of the studied animals. Temporary stress is related to surgery and the development of gas bubbles in the tissues.

Local tissue metabolism influences the corrosion process, a hypothesis suggested by the difference between the corrosion rate of the intramuscular and intraosseous implants. The complex environment of living organisms has a major influence on the corrosion rate of Mg-1Ca alloy. The in vivo corrosion rate varies between a minimum of 0.472 mm/year–0.883 mm/year, intramuscularly and 1.33 mm/year–1.36 mm/year intraosseous. These values are much lower than the in vitro rate which recorded very high average values in SBF, 16.9–28.71 mm/year, and values of 1.98282–6.93987 mm/year in Dulbecco.

In vivo corrosion rates are significantly lower than in vitro rates, regardless of the medium used.

The bioresorbable alloy Mg-1Ca has the ability to osseointegrate, the adjacent bone being directly adherent to the surface of the alloy. At the same time, it has the capacity of osteoconductivity, the bone developing on the surface of the implant. SEM and EDS examinations confirm the biodegradation process of the alloy, but without its disintegration at the established time periods. At the same time, the formation of hydroxyapatite products on the surface of the alloy is observed, with a positive role on its osteoconductivity and osteointegration capacity.

The obtained data are encouraging, justifying the continuation of research through in vivo studies compared to other studies [16,37,38,39].

## 5. Conclusions

This study investigated the corrosion rate and behavior of magnesium alloys during the biodegradation process. The results indicate that local tissue metabolism influences the corrosion process (with a significant difference between muscle corrosion rate and bone corrosion rate). We have shown that In vivo corrosion rate is significantly lower than the In vitro rate. Thus, regarding its behavior, the Mg-1Ca alloy is safe, with a high degree of biocompatibility. The studied Mg-1Ca alloy meets the biocompatibility requirements, being composed of two non-toxic elements, naturally present in the human body, with an important role in bone metabolism. Increased corrosion rate and hydrogen emission during resorption are the main concerns when considering the use of Mg-1Ca alloy in orthopedic medical applications.

The present study tries to answer the controversies in the literature regarding the behavior of magnesium alloys In vitro and In vivo, performing both types of tests on the same type of alloy, in our case Mg-1Ca.

The In vitro tests performed in the present study represent an important and mandatory stage of research on the corrosion and cytotoxicity of the Mg-1Ca alloy, having the advantage of an increased replicability. The Mg-1Ca alloy induces the alkalization of the culture medium during resorption, and the alkaline pH directly influences the corrosion process, with high values allowing the formation and stabilization of the protective layers.

The environment plays a crucial role in the corrosion process; Dulbecco-type culture media with complex composition will induce a decrease in the corrosion rate and the release of hydrogen compared to SBF, and the in vivo corrosion rate is much lower compared to in vitro rate. We also noticed differences in the corrosion rate, depending on the type of tissue in which the alloy is implanted. However, regardless of the media used, we recorded a significantly lower In vivo corrosion rate than the In vitro one.

In order to evaluate the quality of the Mg-1Ca alloy in medical applications, it is necessary to coordinate In vitro studies with In vivo ones, objective results being obtained by performing both types of tests on the same alloy, in the same study, as we did. Thus, the Mg-1Ca alloy represents a promising solution for the future, for orthopedic surgery, presenting an In vivo corrosion resistance clearly superior to that In vitro, without toxic phenomena on the adjacent tissues.

Biodegradation of the Mg-1Ca alloy does not produce local toxic phenomena, bone or muscle tissue in the immediate vicinity of the implant, or gas bubbles, having viability characteristics similar to the control samples. The alloy also has the capacity of osteointegration and osteoconductivity. SEM and EDS examinations confirm the biodegradation process of the alloy, but without its disintegration at the established time periods. At the same time, the formation of hydroxyapatite products on the surface of the alloy was observed, with a positive role on its osteoconductivity and osteointegration capacity. Therefore, the Mg-1Ca alloy represents a promising solution for orthopedic surgery, at the present time being able to find applicability in the small bones: hand or foot.

## Figures and Tables

**Figure 1 materials-14-00084-f001:**
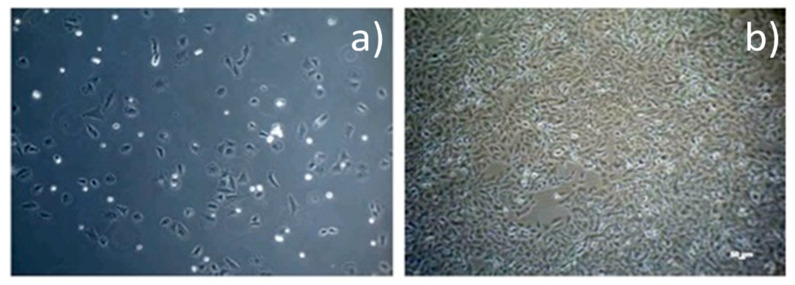
Optical microscopy image of osteoblasts evolution (**a**) day 1, culture medium; (**b**) 80% confluence.

**Figure 2 materials-14-00084-f002:**
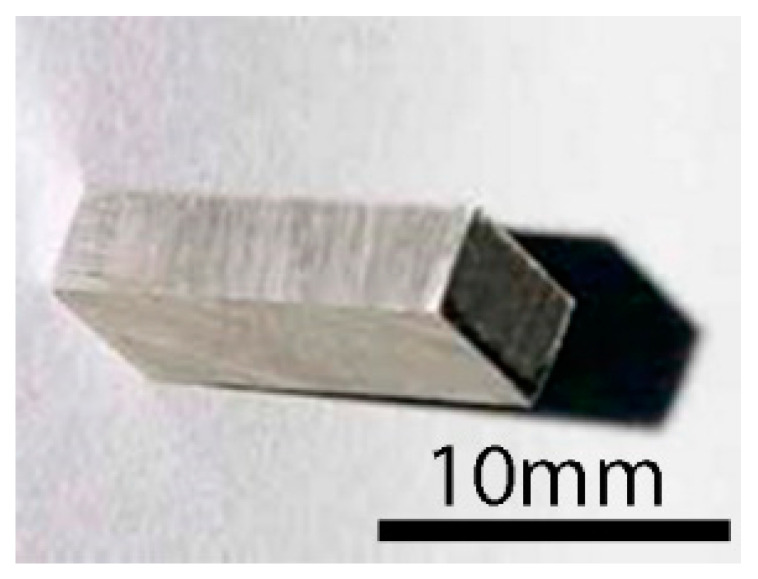
Mg-1Ca sample view.

**Figure 3 materials-14-00084-f003:**
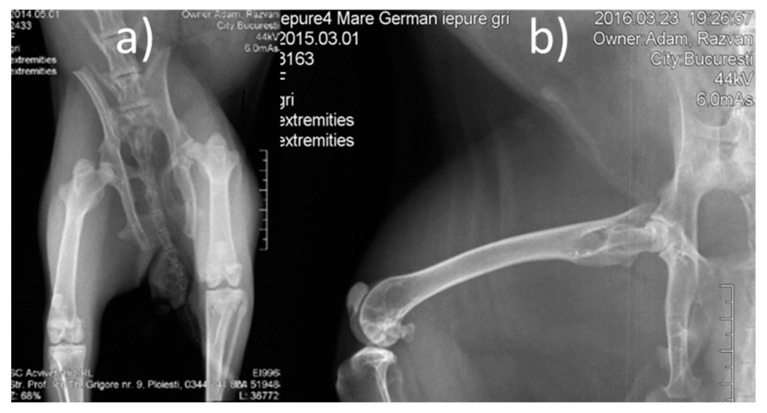
Pre-operative X-ray images (**a**) front view, (**b**) lateral view.

**Figure 4 materials-14-00084-f004:**
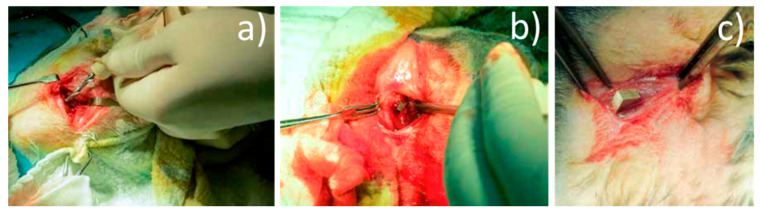
Surgical procedure: (**a**) drill hole, (**b**) implant bone insertion, (**c**) muscle insertion.

**Figure 5 materials-14-00084-f005:**
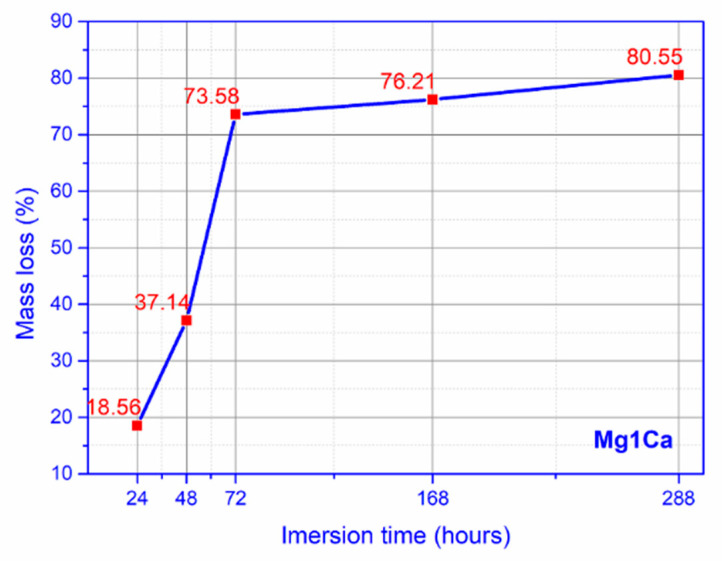
Mass loss simulated biological fluid (SBF) (mean values for three samples).

**Figure 6 materials-14-00084-f006:**
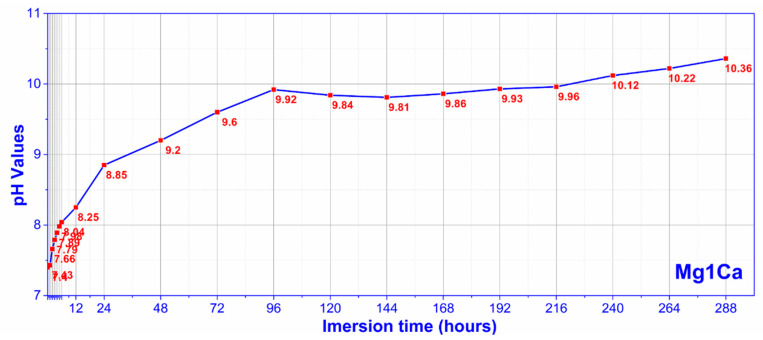
pH evolution in SBF (mean values for three samples).

**Figure 7 materials-14-00084-f007:**
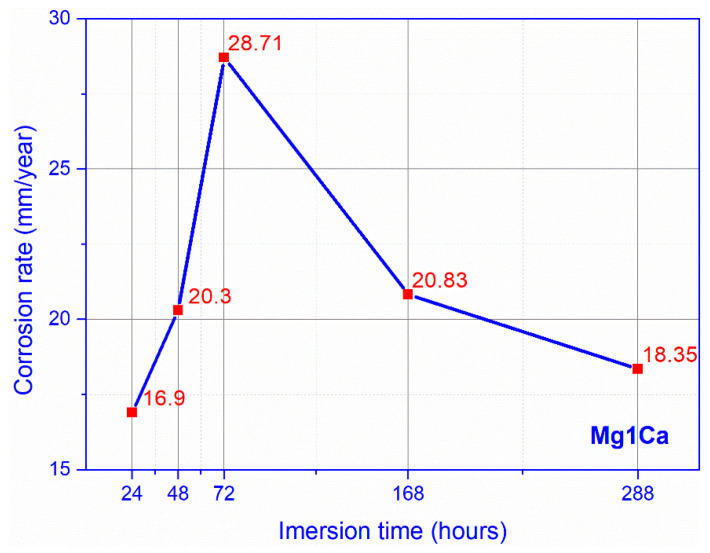
Corrosion rate SBF (mean values for three samples).

**Figure 8 materials-14-00084-f008:**
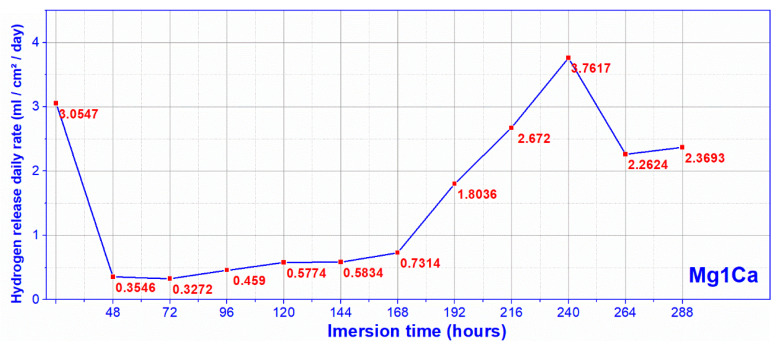
Hydrogen release rate, Dulbecco (mean values for three samples).

**Figure 9 materials-14-00084-f009:**
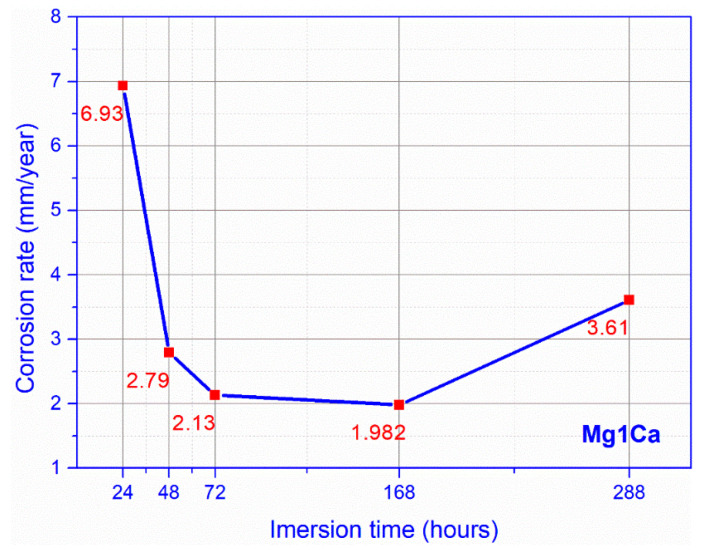
Corrosion rate, Dulbecco (mean values for three samples).

**Figure 10 materials-14-00084-f010:**
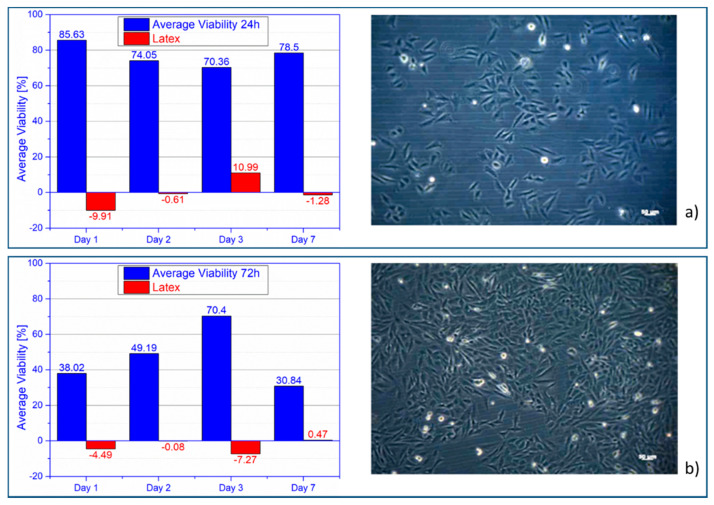
Average cell viability alloy vs. latex (left) and osteoblasts (right): (**a**) 24 h and (**b**) 72 h extract.

**Figure 11 materials-14-00084-f011:**
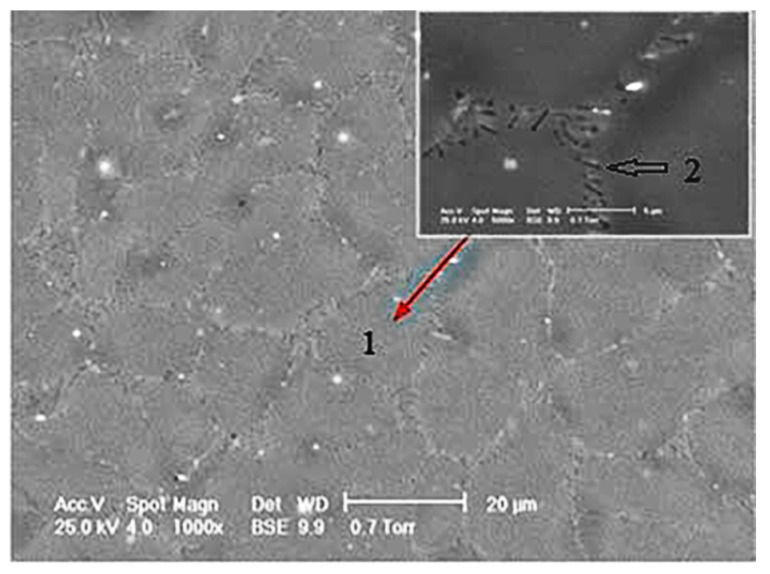
Mg-1Ca SEM image, before immersion.

**Figure 12 materials-14-00084-f012:**
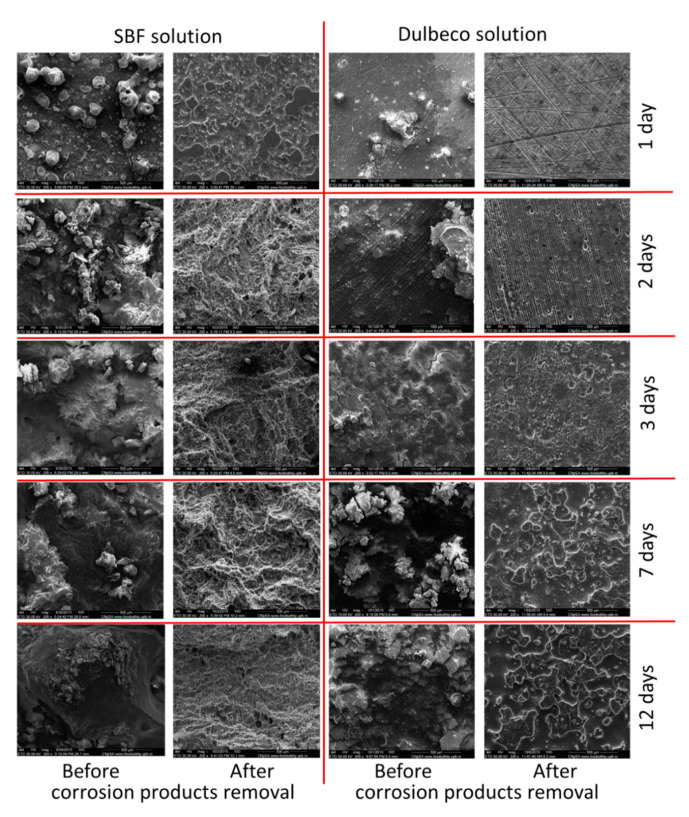
SEM images on the samples surfaces at 1, 2, 3, 7, and 12 days after immersion in SBF and Dulbecco medium before and after the corrosion products removal.

**Figure 13 materials-14-00084-f013:**
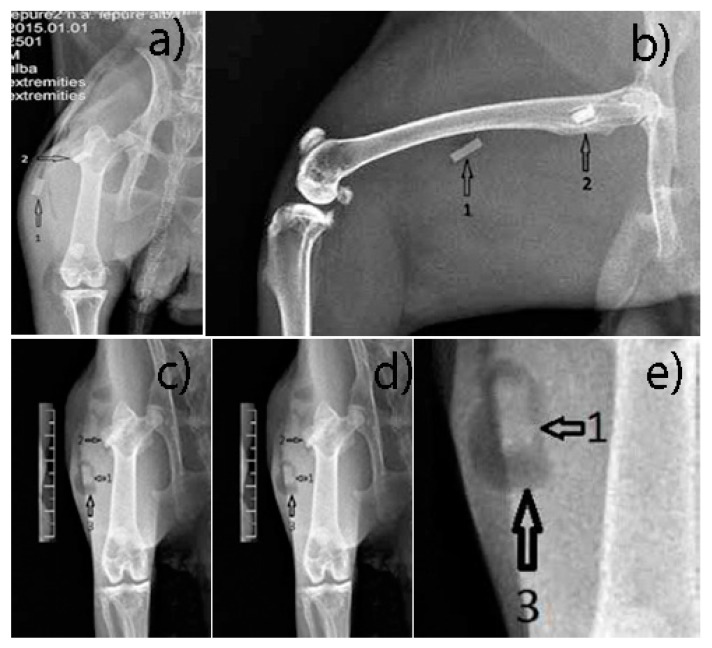
(**a**,**b**) Post-Operative X-ray, (**c**–**e**) 6 weeks post-operative X ray. 1—parallelepiped implant inserted into the thigh muscles; 2—parallelepiped implant inserted into the bone tissue, greater femoral trochanter; 3—hydrogen bubbles formed around the implant.

**Figure 14 materials-14-00084-f014:**
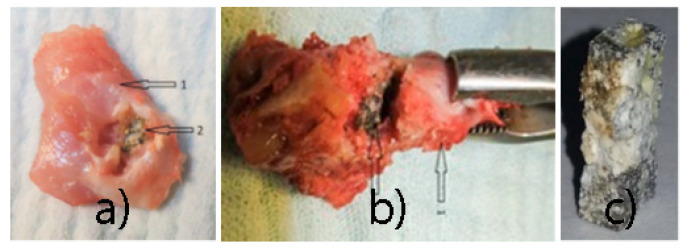
Post-extraction images: (**a**) muscular block (1), alloy sample (2); (**b**) bone block (1), alloy sample (2), (**c**) alloy sample after extraction.

**Figure 15 materials-14-00084-f015:**
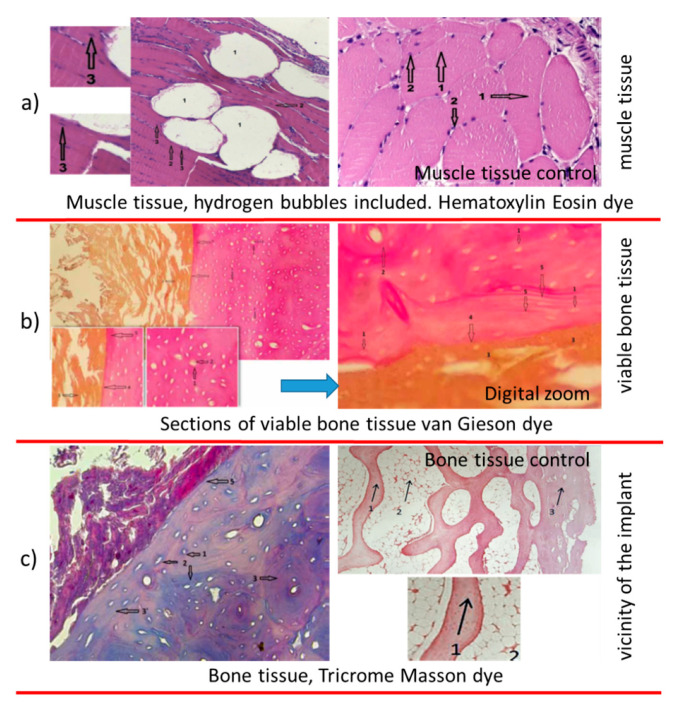
Histological images on different tissue types. (**a**) muscle tissue; (**b**) viable bone tissue; (**c**) bone tissue.

**Figure 16 materials-14-00084-f016:**
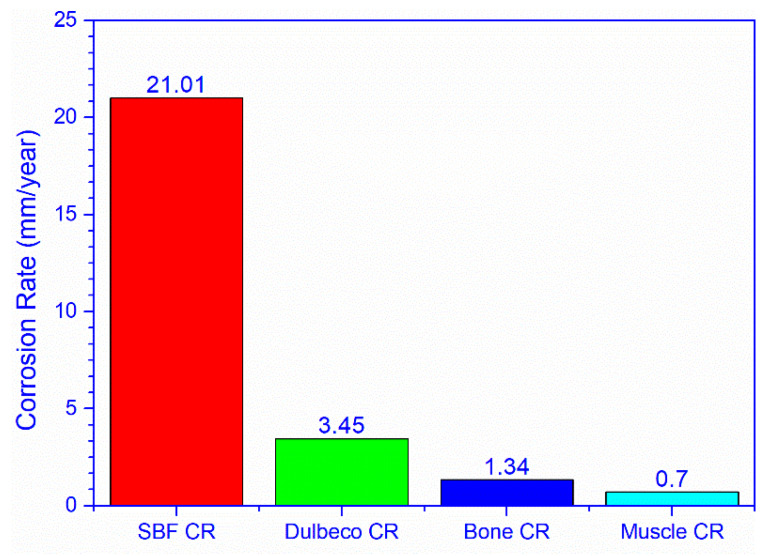
In vitro–in vivo corrosion rate comparison.

**Table 1 materials-14-00084-t001:** SEM-EDS results for the prepared Mg-Ca alloy substrates.

Spectrum	Mg	Ca	Total
1	99.02	0.98	100.0
2	99.00	1.00	100.0
3	98.92	1.08	100.0

All results in weight%.

**Table 2 materials-14-00084-t002:** EDS results before immersion and after 7 days’ examination.

Time	Medium	Corrosion Products Removal	Chemical Elements (wt%)								
			Mg	Ca	Na	P	O	Au	Cl	C	Other
Before immersion	-	-	99	1	-	-	-	-	-	-	-
1 day	SBF solution	Before	86.63	1.07	-	0.83	11.34	0.13	-	-	-
		After	98.85	1.04	-	-	-	0.11	-	-	-
	Dulbecco solution	Before	85.13	2.08	-	2.11	10.65	0.03	-	-	-
		After	95.03	2.01	-	-	2.96	-	-	-	-
2 days	SBF solution	Before	85.65	0.92	-	-	12.88	0.55	-	-	-
		After	93.23	1.14	-	-	5.12	0.51	-	-	-
	Dulbecco solution	Before	83.76	2.29	1.14	2.89	9.01	0.32	0.59	-	-
		After	94.98	1.96	-	-	3.06	-	-	-	-
3 days	SBF solution	Before	84.08	1.32	-	-	14.28	0.32	-	-	-
		After	93.55	1.14	-	-	5.04	0.27	-	-	-
	Dulbecco solution	Before	81.74	2.44	2.06	3.21	9.65	0.23	-	0.67	-
		After	91.11	2.08	-	-	6.81	-	-	-	-
7 days	SBF solution	Before	82.97	1.43	-	-	14.81	0.79	-	-	-
		After	93.79	1.09	-	-	4.93	0.19	-	-	-
	Dulbecco solution	Before	79.78	3.18	-	4.35	11.59	-	-	0.76	0.34
		After	89.31	2.02	-	-	8.67	-	-	-	-
12 days	SBF solution	Before	82.16	1.92	-	-	15.13	0.79	-	-	-
		After	90.39	1.24	-	-	8.26	0.11	-	-	-
	Dulbecco solution	Before	76.03	1.99	2.23	5.89	12.65	0.23	0.55	0.43	-
		After	88.76	1.53	-	-	9.71	-	-	-	-

**Table 3 materials-14-00084-t003:** In vivo corrosion rate.

No	Implant Tissue	Δ Weight (g)	Corrosion Rate (mm/year)
1	Bone	0.0340	1.36
2		0.0352	1.35
3		0.0320	1.33
4	Muscle	0.0228	0.883
5		0.0120	0.472
6		0.0190	0.755

**Table 4 materials-14-00084-t004:** Statistical analysis, corrosion rate (CR) SBF vs. In vivo.

Test Statistics	Corrosion Rate
Mann–Whitney U	0.000
Wilcoxon W	6.000
Z	−2.236
Asymp.Sig. (2-tailed)	0.025
Exact Sig. (2*(1-tailed Sig.))	0.036b

Note: Asymp.Sig.—Asymptotic Significance; Exact Sig.—Exact Significance.

**Table 5 materials-14-00084-t005:** Statistical analysis, CR Dulbecco vs. In vivo.

Test Statistics	Corrosion Rate
Mann–Whitney U	0.000
Wilcoxon W	6.000
Z	−2.324
Asymp.Sig. (2-tailed)	0.020
Exact Sig. (2*(1-tailed Sig.))	0.24b

Note: Asymp.Sig.—Asymptotic Significance; Exact Sig.—Exact Significance.

## Data Availability

Data available on request due to privacy or ethical restrictions. The data presented in this study are available on request from the corresponding author. The data are not publicly available due to privacy reasons.

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
