# Peer review of "Comparative Assessment of In Vitro and In Vivo Biodegradation of Mg-1Ca Magnesium Alloys for Orthopedic Applications"

_materials, 2020, doi:10.3390/ma14010084_

Round 1

Reviewer 1 Report

The results presented are interesting and the paper is complete.

Authors have deal with Mg-Ca alloy, and it is very perspective as 

resorbable biomaterial for small medical implants. 

The main aim of the paper is comparative biotesting. 

Some remarks.

1. Data about producer of Mg-Ca alloy should be added.

2. Description of the Mg-Ca alloy should be added. The element 

composition of alloy, phase composition, the mean value of grain size

could be reported in Section 2. Materials and Methods.

3. The preliminary treatment (cutting, polishing and so on) should be

added in Section 2. Materials and Methods. Moreover, the complete treatment 

of samples for "in vivo" test could be reported.

4. The section 4. Conclusions is short, and it is not informative.

Conclusions should be expanded.

5. Figure 13 is not informative; it is impossible to see individual elements of

the EDS spectra. It is better to present the results in form of table.

6. The last paragraph of Section 1. Introduction should be reformulated. It

could be started with words "The aim of the work is ..."

Author Response

  1. Data about producer of Mg-Ca alloy should be added.
  2. Description of the Mg-Ca alloy should be added. The element composition of alloy, phase composition, the mean value of grain size could be reported in Section 2. Materials and Methods.
  3. The preliminary treatment (cutting, polishing and so on) should be added in Section 2. Materials and Methods. Moreover, the complete treatment of samples for "in vivo" test could be reported.

Response for the 1, 2 and 3 queries

Data were added in a new section Samples materials and preparation method – lines 124-138

  1. The section 4. Conclusions is short, and it is not informative. Conclusions should be expanded.

The conclusion section was replaced and was expanded according your suggestions.

  1. Figure 13 is not informative; it is impossible to see individual elements of the EDS spectra. It is better to present the results in form of table.

The Figure 13 was replaced with a table (new table 2) as suggested

  1. The last paragraph of Section 1. Introduction should be reformulated. It could be started with words "The aim of the work is ..."

The last paragraph of section 1 was reformulated according the reviewer indications see LINES 119-120

Reviewer 2 Report

This paper reports physico-chemical, biological and physiological impacts of Mg-1Ca magnesium alloys used as bioresorbable orthopaedic implants.

To my opinion, the manuscript is clearly written, data reported are complete and the subject is appropriate for publication in Materials.

Author Response

Dear reviewer, thank you very much for the time spent to analyse our paper.

Reviewer 3 Report

The problem considered at work is very important and undertaken for many years by scientists. The article deals with the problem of new magnesium implants  for application in  orthopaedic due to its excellent resorbable properties. The problem is current and very significant in the light of the continuous increase in the use of electronic devices. There are science works which try to prepare and test materials for orthopaedic surgery. The authors place their hope in  Mg-1Ca alloy due to it biocompatibility and appropriate corrosion and physicochemical properties.

Experimental tests, as i.e. corrosion, physicochemical and biological properties were performed. The in vitro and in vivo tests were performed which is of great significance.

The manuscript is interesting. However, there are several points that I would like to address:

- the hypothesis should be clearly written,

- line 41: “the se of magnesium implants”  - please correct in whole manuscript

- line 104: in vitro, in vivo – should be written in Italic- please correct in whole manuscript

- line 99: ions should be written with superscripts and subscripts - please correct in whole manuscript

-line 116: cm2 / cm3 – please correct in whole manuscript

- mass: should be written as 0.512 g not “grams”

- line 144: “hydrogen”

- CR: mm/years or mm/year – please check which is correct

- tables 2,3 – should be prepared once more as the quality is no enough

- discussion of the experimental results should be more wide, as now it only contains a description of the results without comparison to other materials and other works,

- the list of references is not homogenous – titles of journals are written sometimes with full name and sometimes as abbreviations – please homogenize.

Author Response

- the hypothesis should be clearly written,

A clearer description was added see lines 104-118

- line 41: “the se of magnesium implants”  - please correct in whole manuscript

The error was corrected throughout the whole manuscript

- line 104: in vitro, in vivo – should be written in Italic- please correct in whole manuscript

We have corrected the entire manuscript.

- line 99: ions should be written with superscripts and subscripts - please correct in whole manuscript

We have corrected the entire manuscript.

-line 116: cm/ cm– please correct in whole manuscript

- mass: should be written as 0.512 g not “grams”

We corrected the error.

- line 144: “hydrogen”

We corrected the typing error .

- CR: mm/years or mm/year – please check which is correct

We replaced mm/years with mm/year

- tables 2,3 – should be prepared once more as the quality is no enough

Tables 2 and 3 were replaced – new tables 4 and 5 – lines 392-398

- discussion of the experimental results should be more wide, as now it only contains a description of the results without comparison to other materials and other works,

A wider discussion section was added referring to the experimental results, as suggested – see lines 410-493

- the list of references is not homogenous – titles of journals are written sometimes with full name and sometimes as abbreviations – please homogenize.

The reference list was corrected

Round 2

Reviewer 3 Report

Dear Authors, 

all my comments were taken into consideration and corrected.

Only please correct in line 109: should be "pH"